# Regulation of replication timing in *Saccharomyces cerevisiae*

**Rosie Berners-Lee**[1], **Eamonn Gilmore** [2], **Francisco Berkemeier**[2,3]*, **Michael A. Boemo** [2,3]*

**1** University of St. Andrews, St. Andrews, Fife, United Kingdom, **2** Department of Pathology, University of Cambridge, Cambridge, United Kingdom, **3** Department of Genetics, University of Cambridge, Cambridge, United Kingdom

* fp409@cam.ac.uk (FB); mb915@cam.ac.uk (MAB)

## Abstract

In order to maintain genomic integrity, DNA replication must be highly coordinated. Disruptions in this process can cause replication stress which is aberrant in many pathologies including cancer. Despite this, little is known about the mechanisms governing the temporal regulation of DNA replication initiation, thought to be related to the limited copy number of firing factors. Here, we present a high (1-kilobase) resolution stochastic model of *Saccharomyces cerevisiae* whole-genome replication in which origins compete to associate with limited firing factors. After developing an algorithm to fit this model to replication timing data, we validated the model by reproducing experimental inter-origin distances, origin efficiencies, and replication fork directionality. This suggests the model accurately simulates the aspects of DNA replication most important for determining its dynamics. We also use the model to predict measures of DNA replication dynamics which are yet to be determined experimentally and investigate the potential impacts of variations in firing factor concentrations on DNA replication.

## Author summary

Each time a cell divides, it is essential that its entire genome is copied accurately. The timing of this DNA replication must be highly coordinated, and disruptions to this process are common in many diseases, including cancer. Despite its vital importance, what controls this coordination is still not fully understood. One idea is that the potential sites where replication can start must compete with each other for a limited supply of essential proteins. To explore this, we created a computational model that simulates whole-genome replication in budding yeast. In our model, potential starting points for replication compete to bind with a limited number of essential proteins. We show that the model can reproduce known features of DNA replication dynamics and also predict aspects of the process that have not yet been measured experimentally. By making the model as simple as possible while still capturing the key features of DNA replication,

**Data availability statement:** Models were simulated using Beacon Calculus version 1.1.0

(the latest available at the time of writing) which is available at https://github.com/MBoemo/bcs. Models, fitting algorithms, and analysis scripts are available at https://github.com/rb2065/S_cerevisiae_DNA_rep_model.

**Funding:** This work was supported by the Leverhulme Trust (RPG-2022-028 to MAB). This Leverhulme Trust Research Project Grant included a salary for FB. The funders had no role in the study design, data collection and analysis, decision to publish, or preparation of the manuscript.

**Competing interests:** The authors have declared that no competing interests exist.

we identify the factors most important for determining replication timing. Our model provides a useful tool for investigating how replication is coordinated and may help to guide future research.

## Introduction

To maintain genomic integrity, DNA replication must be tightly controlled to ensure that the entire genome is replicated precisely once per cell cycle [1]. In eukaryotes, replication initiates from multiple sites across the genome known as origins of replication. While the assembly and major enzymatic activity of the replisome is conserved across eukaryotes, there is considerable variability between species as to what genomic features cause loci to act as origins of replication. In *Saccharomyces cerevisiae*, origins occur at defined sequences known as autonomously replicating sequences (ARS) [2]. This, along with their short cell cycle and simple, well-understood genome [3], makes *S. cerevisiae* a good model organism for studying DNA replication. Each ARS shares a common ARS consensus sequence (ACS), which is essential but not sufficient for replication initiation. Local chromatin architecture and nucleosome positioning are also thought to influence the locations of origins [4,5]. The locations of 829 *S. cerevisiae* origins have been mapped and are available in OriDB [6].

As depicted in Fig 1A, DNA replication initiation at origins is divided into two temporally distinct stages: licensing and firing (reviewed in [7]). Licensing involves loading of the core of the replicative helicase, the MCM2-7 complex, onto replication origins and occurs

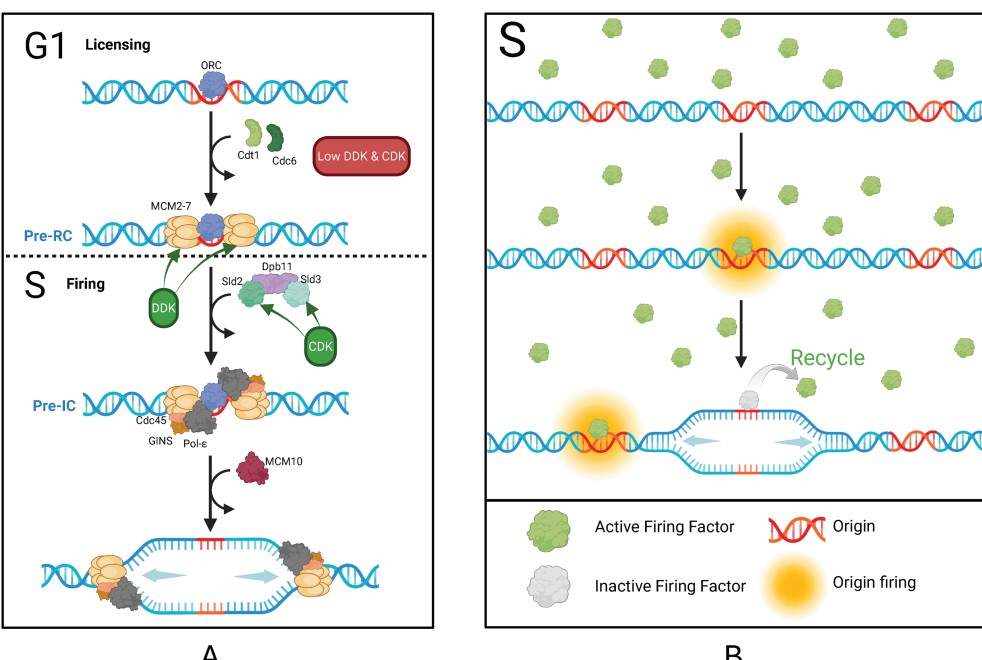

**Fig 1. Initiation of DNA replication.** A: Mechanism of DNA replication initiation at origins. Licensing occurs during G1 at low levels of DDK and CDK activity. Firing occurs during S phase at high levels of DDK and CDK activity. B: DNA replication initiation as represented in our model. Our model simplifies DNA replication initiation to the S phase, representing all necessary proteins for origin firing as a single "firing factor". This factor binds to origins with different affinities and is subsequently recycled for future use. Created in BioRender.com.

at low levels of cyclin-dependent kinase (CDK) and Dbf4-dependent kinase (DDK) activity during late-M and G1 phases of the cell cycle. The process begins with the origin recognition complex (ORC) binding to origins in an ATP-dependent manner. Subsequent recruitment of Cdc6 and MCM2-7/Cdt1 proteins facilitates the loading of a head-to-head double-hexamer of the MCM2-7 replicative helicase around the duplex origin DNA. Importantly, in this form, the MCM2-7 complex is inactive. The resulting pre-replication complex (pre-RC) licences the origin for replication [8]. The transition from G1 to S phase is accompanied by increased CDK and DDK activity that facilitates origin firing, which involves conversion of the inactive MCM2-7 double hexamer to two active CMG helicases [9]. Increased CDK activity also inhibits further licensing, thereby ensuring only one round of replication per cell cycle. The process of origin firing is driven by the coordinated action of multiple proteins, collectively referred to as "firing factors". DDK-mediated phosphorylation of the pre-RC drives recruitment of the Sld7-Sld3 complex and Cdc45 [10]. Away from the DNA, Sld2 binds to GINS and Pol-ε. Phosphorylation of Sld2 and Sld3 by CDK allows both to bind to Dpb11, bringing together their associated proteins at the origin to form the pre-initiation complex (pre-IC) [11]. Sld2, Sld3, and Dpb11 then depart, leaving the remaining CMG helicase (Cdc45, MCM2-7, GINS) and pol-ε. Subsequent binding of MCM10, along with other replication factors, results in origin firing by allowing rearrangements of CMG and origin DNA to form the active replisome [12]. DNA synthesis then progresses bidirectionally from each fired origin, with dNTPs continuously incorporated into the growing nascent DNA strands. Replication forks continue advancing until they are terminated by either converging with opposing forks or reaching the ends of chromosomes [13].

The essential firing factors Sld2, Sld3, Cdc45, Dpb11, and the DDK subunit Dbf4 have all been found to be in low abundance in *S. cerevisiae*. Overexpression of these leads to premature firing of late origins [14,15]. An origin's affinity for firing factors is influenced by multiple factors including chromatin accessibility and nuclear localisation [12,16,17]. Stochastic firing means that, whilst the replication timing profile is reproducible at a population level, the set of origins that fire and their precise firing times differ between cells and cell cycles [18–20]. Therefore, single-molecule methods such as DNA fibre analysis are needed to study details of DNA replication which are smoothed out in population averaging obtained from bulk experiments [21,22]. However, these techniques tend to have low throughput, and while recent long-read sequencing [23–25] and optical mapping [19] techniques have expanded the toolkit to study DNA replication, mathematical models are a valuable complement to these techniques to study DNA replication dynamics.

Previous work on mathematical modelling of DNA replication in *S. cerevisiae* has employed analytical models based on the Kolmogorov-Johnson-Mehl-Avrami equation to derive local, time-dependent firing rates, capturing genome-wide stochastic initiation patterns [26,27]. Deterministic models have also used origin positions and fork migration rates to predict replication timing profiles [28], while other studies have highlighted how stochastic origin usage enhances replication robustness [29]. Additionally, other models characterised origin firing on chromosome 6 by defining each origin's competence, reflecting firing probability through pre-RC assembly, and a Gaussian-distributed activation time [30,31]. Whilst fast to compute, none of these modelling approaches includes firing factors. Rate-limiting firing factors were incorporated into a stochastic model by [32] in which firing factors were loaded into the system gradually and traveled with the replication forks, being released upon fork termination. While this is thought to be the case for Cdc45 [15], it does not account for other firing factors. Stochastically varying fork speeds were incorporated into the model by [33]. However, this contrasts with data from [25] which showed a consistent fork speed

across the budding yeast genome. A Bayesian algorithm was used by [34] to infer the origin firing time distributions of *S. cerevisiae* chromosome 10 from Okazaki fragment analysis. However, this model only considered three consecutive origins at a time and used "replicated base pairs" rather than conventional time units. A more complex model developed by [35] attempted to describe all of the core processes governing DNA replication initiation in *S. cerevisiae*. However, the challenge of incorporating every relevant factor restricted their model to capturing the kinetics of early, but not late, firing origin.

Whilst a broad range of different measures of DNA replication dynamics have been investigated across the myriad of existing models, differences in assumptions and generalisations between these models make it difficult to compare their findings. Furthermore, many rely on complex, rigid mathematical equations, making them difficult to interpret and modify. To gain a better understanding of the core mechanisms governing DNA replication dynamics, there is therefore still a need for a DNA replication model which is as simple as possible whilst still capturing the main features and from which many aspects of DNA replication dynamics can be extracted. Here, we address this by presenting a stochastic model for *S. cerevisiae* whole-genome DNA replication in which origins compete to associate with limited firing factors, needed for initiation, which then recycle to be used again (Fig 1B) [14]. We constructed this model using Beacon Calculus, a process algebra designed for modelling biological systems [36]. The inherent stochasticity of Beacon Calculus makes it well suited to modelling DNA replication and its simplicity makes it intuitive, straightforward to interpret, and easily modifiable. The detailed output of Beacon Calculus simulations allows DNA replication to be investigated at a single-molecule level and enables multiple features of its dynamics to be extracted from the same output for more comprehensive insights.

## Materials and methods

### The simulation

**Origin positions.**   We obtained the positions of origins from OriDB [6], which catalogues 829 origins, classifying them as 'Confirmed' (410), 'Likely' (216), or 'Dubious' (203). In our model, we only include 'Confirmed' origins (verified by ARS assays and/or 2D gel analysis) and 'Likely' origins (identified in two or more microarray studies). We excluded 'Dubious' origins, as these were identified by only a single microarray study.

**Model formulation.**   We created a stochastic model for *S. cerevisiae* whole-genome replication in which origins compete to associate with limited firing factors, needed for activation, which then recycle to be used again (Fig 1B). To construct our model, we used Beacon Calculus, a process algebra designed for modelling biological systems [36] (S1 Fig). In Beacon Calculus, each component of the system is represented as a process capable of performing one or more actions. The Beacon Calculus model is simulated according to a modified Gillespie algorithm, with each action being associated with a rate that represents the parameter of an exponential distribution. This makes models written in the Beacon Calculus inherently stochastic and therefore well-suited to modelling DNA replication.

In our model, each origin is characterised by four parameters: its chromosome number, `[ch]`; position on the chromosome `[i]`; chromosome length `[length]`; and the rate at which it associates with firing factors, `[fire]`. The model was formulated such that chromosomal positions had a spatial resolution of one kilobase (kb). Once an origin associates with a firing factor, it initiates two replication forks which move bidirectionally, replicating DNA at a constant rate of 1.4 kb/min [33,37]. Forks terminate when they reach either the end of the chromosome or an oncoming replication fork [13]. In our model, an origin is passively replicated and therefore removed from the system when a fork moves through its position before

it can fire. After associating with an origin, firing factors are temporarily inactive while they are recycled for reuse. We assume that firing factors are available immediately following the transition from G1 to S phase [9]. Therefore, all firing factors are present from the beginning of the simulation.

Each simulation of the model can be interpreted as a single S-phase within an individual cell. Given the stochastic nature of the model, each simulation yields variable results, similar to the variability observed in individual cells [18]. To capture a representative overview of DNA replication dynamics at the population level, unless otherwise stated, we based our analysis on results averaged from 500 simulations of each model.

**Fitting the origin firing rates.**   The model was fitted to experimental replication timing data iteratively through cycles of model prediction, comparing the predicted and experimental replication timings, and adjusting the firing rate to better fit the replication timing data. The replication timing data used was sourced from [38] who used deep sequencing and fluorescence-activated cell sorting (FACS) to determine population-averaged DNA replication timings in asynchronous diploid *S. cerevisiae*. To facilitate comparison with the simulated replication timings from our model, linear interpolation was applied to estimate replication timings for each genomic locus with 1-kb resolution. For iteration $n$ and origin $i$, we denote the origin's firing rate as $f_{i,n}$. For the initial iteration $n = 0$, the firing rate at each origin was calculated using the experimentally determined replication timing at that origin [38], $T_i$, and the number of firing factors in the model, $F$, according to

$$f_{i,0} = \frac{1}{F}\frac{1}{T_i} \tag{1}$$

For subsequent iterations, $n + 1$, new firing rates at each origin, $f_{i,n+1}$, were calculated based on the previous firing rate, $f_{i,n}$, and a power ($\alpha$) of the ratio of its simulated replication timing, $\tilde{T}_{i,n}$, to its experimental replication timing:

$$f_{i,n+1} = f_{i,n}\left(\frac{\tilde{T}_{i,n}}{T_i}\right)^{\alpha} \tag{2}$$

The parameter $\alpha$ was set to 1.2 to achieve efficient fitting by balancing faster fitting times against the risk of taking large steps that cause instability. While there are alternative fitting methods for unreplicated DNA fractions as discussed in [26,39,40], this method is sufficient to capture the prevailing firing rate distribution for given replication timing profiles in *S. cerevisiae*.

**Total error.**   The total error in replication timing was quantified using the Mean Absolute Error (MAE), which was calculated as the mean absolute difference between our simulated replication timing and the experimental data from [38] for each genomic locus with 1-kb resolution. The model was fit for 15 iterations, chosen because the MAE's rate of change approached zero beyond this point indicating that further iterations would not substantially improve the fit. The model with the lowest MAE over the fitting iterations was selected as the final model used for the results in this paper.

**Model configuration.**   The essential firing factor identified to have the lowest copy number is Dpb11, estimated at 200 [15]. For simplicity, we assume that Dpb11 is rate-limiting and therefore set $F = 200$. While the recycling rate is more difficult to estimate, this parameter was set conservatively to 0.05 such that the expected recycling time would be approximately one-third of S phase. While the model's robustness meant that it was also able to be fitted using a range of different combinations of firing factor copy numbers and recycling rates, all analysis

in this study was based on the fitted model configured with $F = 200$ and a recycling rate of 0.05 (Table 1).

## Software and processing

Models were simulated using Beacon Calculus version 1.1.0 (the latest available at the time of writing) https://github.com/MBoemo/bcs, [36]. Models, fitting algorithms, and analysis scripts are available at https://github.com/rb2065/S_cerevisiae_DNA_rep_model.

## Results

### Model convergence

The replication timing predicted by the model converged with the replication timing data from [38] over successive iterations to produce a close match (Fig 2). Fig 2A shows how the model's prediction of replication timing compares to data when using the selected model with the lowest MAE against the data over 15 iterations. Fig 2B demonstrates how our initial predictions of origin firing rates were improved upon by successive fitting iterations. As the iterations progressed, we observed the MAE converging to a minimum value of 1.32 minutes by the 15$^{th}$ iteration (as shown in Fig 2C). The root mean squared error (RMSE) was 1.81 minutes and the $R^2$ value obtained from comparing the experimentally determined and simulated replication timings was 0.94.

**Length of S phase.** In our model, S phase duration is defined by the time required, from the beginning of the simulation, to replicate the entire genome. The mean simulated length of S phase was 93.7 ± 10.8 minutes (Fig 3B) which aligns with time course data from [38]. The low variation in simulated S phase duration shows that, despite its inherent stochasticity, our model consistently achieves DNA replication completion within a biologically realistic timeframe. Moreover, consistent with experimental results, our model indicates that the majority of DNA replication occurs within the first 60 minutes of S phase (Fig 3A).

### Model validation

To validate the model, features of DNA replication dynamics were calculated from the output of 500 model simulations and compared to experimental data that was not used to fit the model.

**Inter-origin distances.** Inter-origin distance (IOD) is defined as the length of DNA between successive origins which fire during the same cell cycle. The mean IOD determined from the model simulation was 57.4 ± 30.9 kb (Fig 3C) and was relatively consistent across chromosomes. This aligns closely with the mean IOD of 55.6 ± 30.3 kb which has been previously established by DNA combing experiments [41].

**Table 1. Summary of model parameters.** Summary of the main parameters used in the model and the rationale for their chosen values.

| Model Parameter | Value | Rationale |
|---|---|---|
| Fork speed (kb/min) | 1.4 | Determined experimentally for chromosome 6 by [37] and found to be optimum for modelling by [33]. |
| Number of firing factors | 200 | The copy number of Dpb11 [15]. |
| Recycling rate | 0.05 | Selected to take approximately one third of S phase. |
| Number of origins | 626 | Locations of 'confirmed' and 'likely' origins from OriDB [6]. |

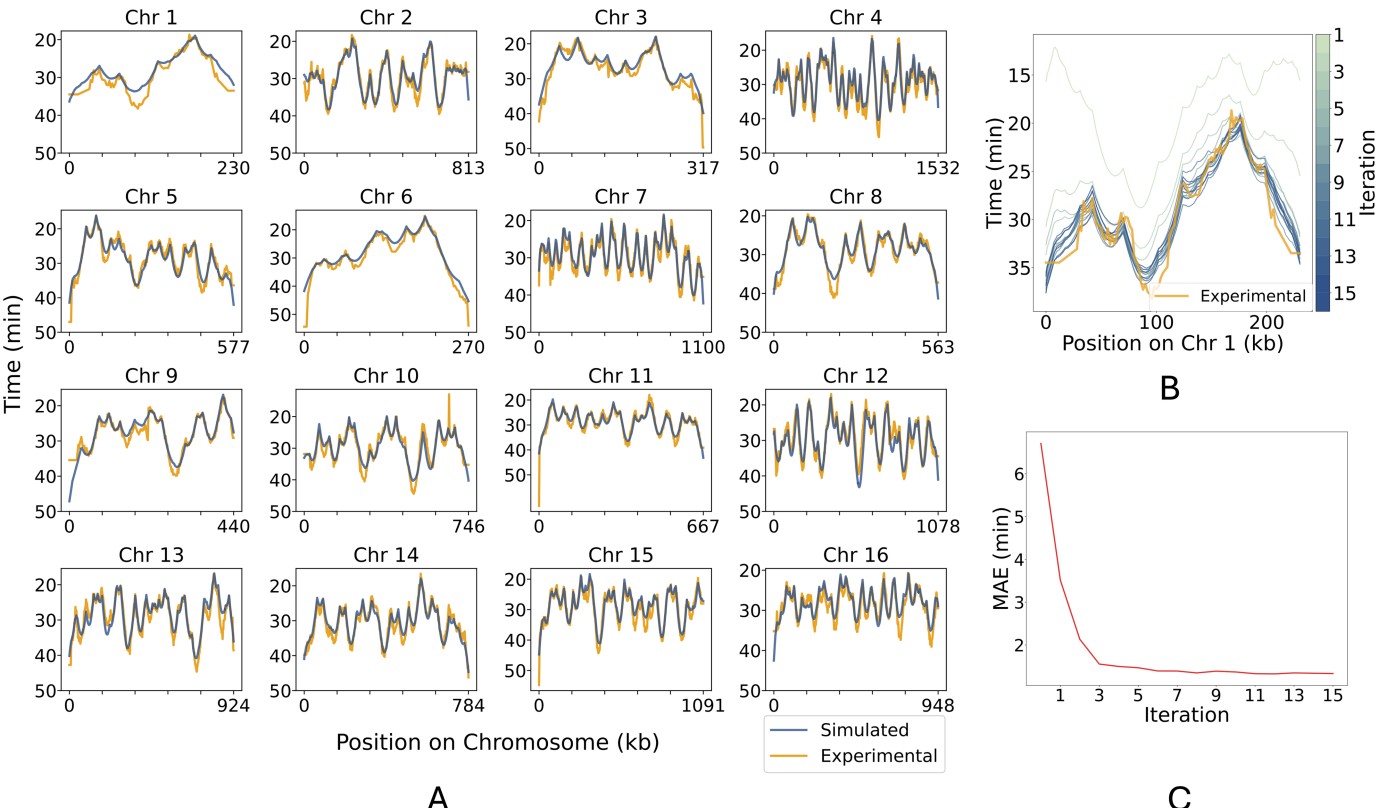

**Fig 2. Fitting the model to replication timing data.** A: Replication timing profiles from simulations using the fitted model (*blue*) and experimental data from [38] (*orange*). B: Replication timing profile of chromosome 1 over the fitting, highlighting the incremental convergence towards the real timing profile. Model fitting iterations are represented as a colour gradient from *light green* to *blue*. Experimental data (*orange*) is also shown for comparison. C: The change in Mean Absolute Error (MAE) in replication timing over the fitting iterations.

**Origin efficiency.** Origin efficiencies were calculated as the percentage of simulations in which each origin fired. As expected, more efficient origins tended to have earlier replication timings. The efficiencies of 433 of the 626 origins included our model have been previously determined experimentally from Okazaki fragment analysis [42]. The mean absolute difference between the experimental and simulated origin efficiencies was 18.7% ± 14%. The mean signed difference of 6.2% ± 22% is close to zero and this was relatively consistent across all chromosomes (Fig 3D). This indicates that our model's estimation of origin efficiencies is not markedly skewed towards either overestimation or underestimation. The large variation in these differences between simulated and experimental origin efficiencies is comparable to the variation observed when comparing efficiencies derived from Okazaki fragment analysis and replication timing data [38,42]. This suggests that the discrepancies between our model and experimental data are within the expected range of variability.

**Replication fork directionality.** Replication fork directionality (RFD) is a measure of the proportion of cell cycles in which a particular position is replicated by either the leftward or rightward moving fork. RFD ranges from -1 to 1 where a RFD of -1 corresponds to positions which are always replicated by leftward moving replication forks whilst positions exclusively replicated by rightward moving replication forks have a RFD of 1. [43] have used Okazaki fragment sequencing (OK-seq) [44] to determine the RFD for the entire *S. cerevisiae* genome.

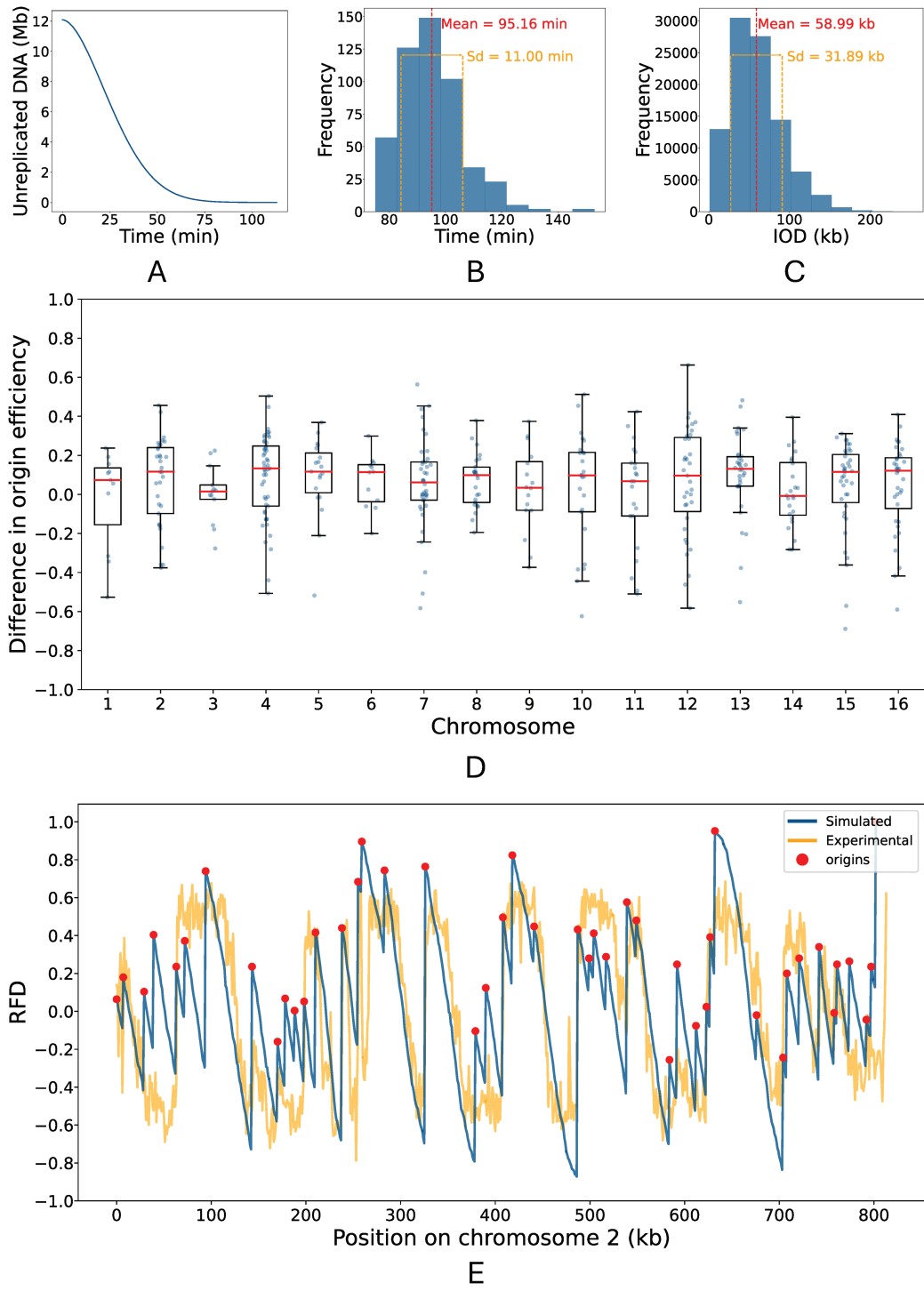

**Fig 3. Comparing simulated and experimental replication dynamics.** A: Length of unreplicated DNA over time. B: Distribution of the total time taken to complete DNA replication (lengths of S phase). C: Distribution of inter-origin distances (IODs). The mean (*red*) and standard deviation (*orange*) are shown. D: Box plots of the signed difference between simulated and experimentally determined origin efficiency across each Chromosome. Efficiency differences for individual origins are shown as scatter points (*blue*). E: Simulated (*blue*) and experimentally determined (*orange*) replication fork directionality (RFD) for chromosome 2. Positions of origins are shown as *red points*. RFD represents the proportion leftward and rightward moving forks which replicate each position, ranging from -1 (exclusively leftward moving forks) to 1 (exclusively rightward forks). All plots were derived from 500 simulation a version of the model in which *F* =200 and the recycling rate was 0.05.

As demonstrated for chromosome 2 in Fig 3E, the simulated RFD from our model was in relatively close agreement with this experimentally determined RFD. The mean percentage absolute difference between these experimentally determined RFDs and those determined by our model was 19.9% ± 16.0%. This difference can be partly attributed to the increased noise in the experimental RFD.

## Model predictions

Our model is also capable of predicting various aspects of DNA replication that have not yet been fully explored experimentally. While this means these predictions cannot currently be verified by experiments, they offer valuable insights into the dynamics of DNA replication and suggest directions for future research.

**Active replication forks.** Our model predicts the mean number of replication forks which are active at each time point over the simulated S phase (Fig 4A). This reached a maximum of 200 active replication forks at 22 minutes into S phase.

**Distribution of origin firing times.** From our model, the distribution of firing times for each origin can be extracted. Using a subset of chromosome 2 origins as an example, Fig 4B illustrates how firing time distributions correspond to origin efficiencies. This is shown for all chromosome 2 origins in S2 Fig. More efficient origins do not necessarily fire earlier or have less varied firing time distributions. Given the limited components in our minimal model, this suggests positioning and context among other origins are more important than efficiency in determining firing time which is consistent with the discussion in [45]. Additionally, many origins exhibited weakly bimodal firing time distributions, which also did not correlate with their efficiencies.

**Replicons.** Our model's capacity to track individual replication forks also enables the investigation of replicons, defined as the segment of DNA replicated from a single origin. Replicon length is an important metric for investigating DNA replication dynamics because, as well as the efficiency of the origin in question, it is also influenced by its proximity to neighboring origins and their respective efficiencies. From our model, we calculated the average replicon length for each origin. Fig 4C exemplifies this for chromosome 2. As expected, there was a strong positive correlation between an origin's efficiency and its average replicon length. Although [46] have used the single molecule technique Replicon-seq to study replicons in *S. cerevisiae* experimentally, their focus on early S phase and the potential bias of Replicon-seq towards detecting shorter replicons limits its use for validating our model. Furthermore, by determining replicon lengths beyond just the early stages of replication, our model provides a broader overview which captures the dynamics of both early and late firing origins.

**Variation in replication timing.** Our model enables us to predict variability in replication timing, as shown for chromosome 2 in Fig 4D. Notably, we observed local maxima in variability at the boundaries of origin clusters. The average standard deviation for replication timings at the origins stood at 16.82 ± 2.36 minutes whereas the average standard deviation for replication timings across the entire genome was slightly lower, at 15.50 ± 2.14 minutes.

## Influence of firing factors on DNA replication

**Impact of firing factor availability.** In our model, the rate at which origins fire is determined by the interplay between three factors: the affinity of individual origins for firing factors, the abundance of "free" firing factors able to associate with origins, and the number of available origins yet to fire or be passively replicated. This dynamic, averaged over 500

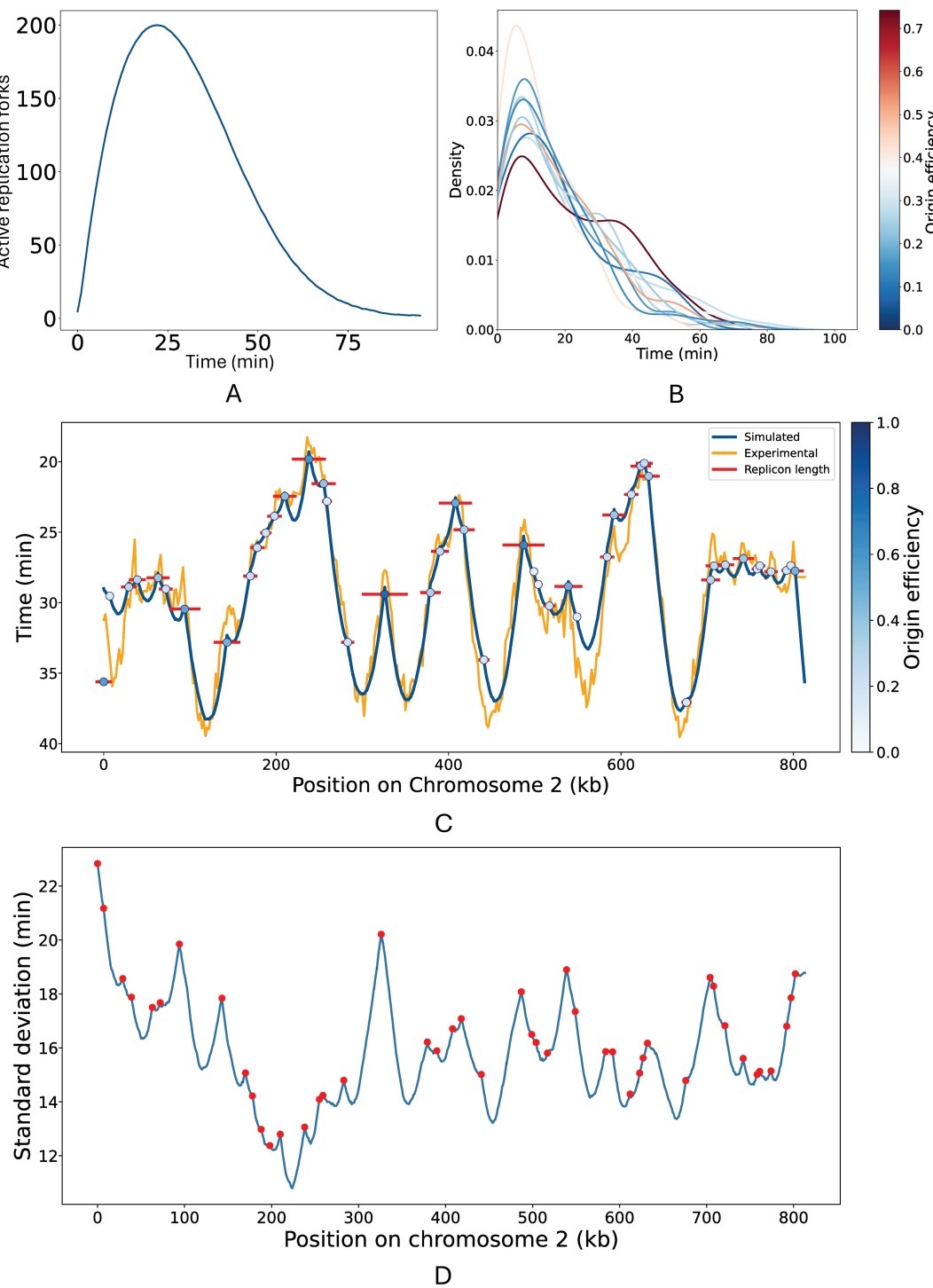

**Fig 4. Predictions of DNA replication dynamics.** A: The number of active replication forks over time. These plots were derived from a version of the model which used $F = 200$ and a recycling rate of 0.05. B: Kernel density plot showing the distribution of firing times for origin on chromosome 2. Each line represents a separate origin, with the shade colour gradient from *blue* to *red* reflecting its efficiency. For improved visualisation, only 1 in 5 origins, ordered by efficiency, are shown. C: Chromosome 2 replication dynamics. Simulated and experimental replication timing profiles are shown in *blue* and *orange* respectively. Origins are shown as *circles* with the shade of *blue* reflecting their efficiencies. The average replicon length of each origin is shown in *red*. D: The standard deviation of simulated replication timing for each kb on chromosome 2. The positions of origins are shown in *red*. All plots were derived from 500 simulation a version of the model in which $F = 200$ and the recycling rate was 0.05.

simulations, is illustrated in Fig 5A. The simulation begins with maximal availability of firing factors and origins, causing high initial origin firing. This initial swell of activity reduces the pool of both free firing factors and available origins, subsequently leading to a decline in the rate of origin firing. A minimum of 122.84 free firing factors, reached at 21 minutes into S phase, is closely followed by the peak in replication fork activity (Fig 4A). As the simulation progresses, firing factors are recycled back into the available pool. This increases the probability of some of the few remaining origins firing, thereby facilitating the timely completion of genome replication within the duration of the S phase. Of the 626 origins included our model, a mean of only 206.43 ± 6.72 origins fire per simulation.

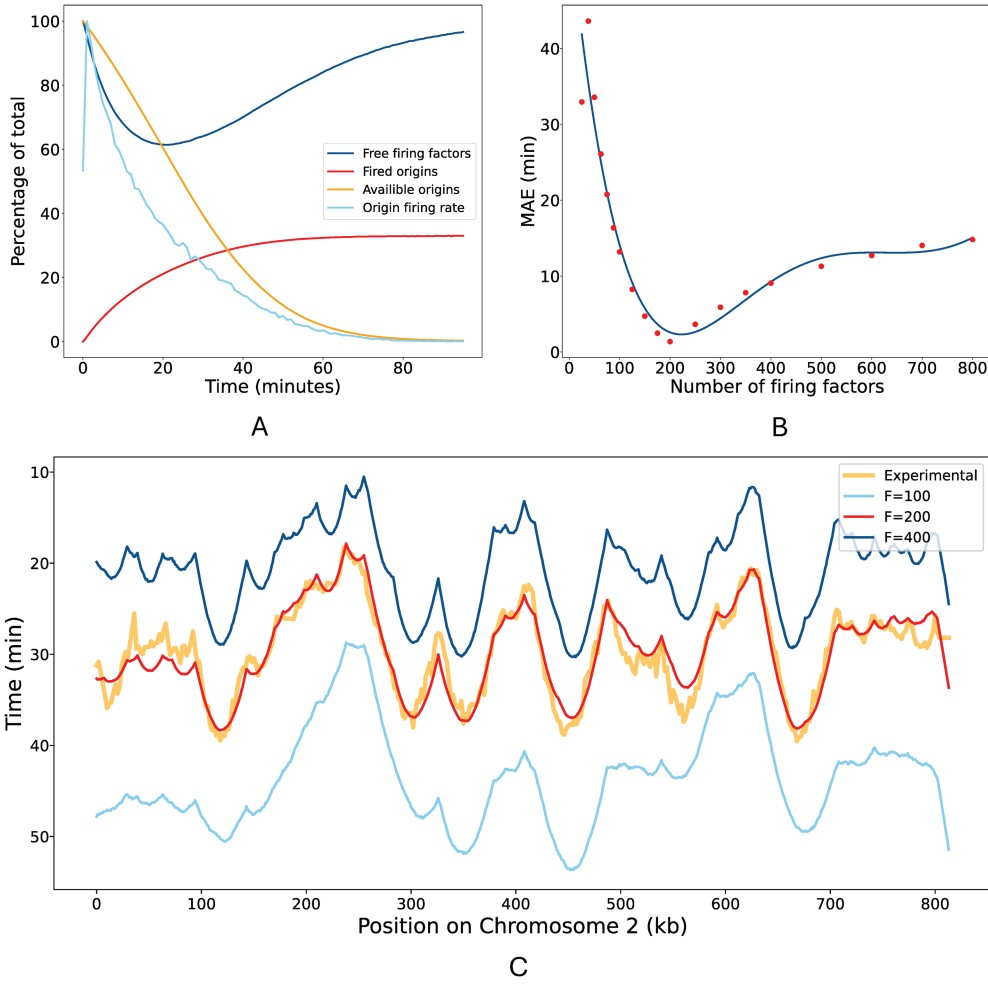

A

B

C

**Fig 5. Influence of firing factor availability on origin firing dynamics.** A: The percentage of active firing factors (*dark blue*), fired origins (*orange*), origins available to fire (*red*) and rate of origin firing over time (*light blue*). Here, available origins refers to origins which have not already either fired or been passively replicated. For this plot, the time has been cropped at the mean simulation length. B: Mean Absolute Error (MAE) in replication timing with different numbers of firing factors. Data points from simulations (*red*) have been fit to a fifth-degree polynomial (*blue*). C: Simulated replication timing profiles for chromosome 2 from models using F = 100 (*light blue*), F = 200 (*red*), and F = 400 (*dark blue*). In all models, origin firing fates were fitted using F = 200. The experimentally determined replication timing profile is also shown for comparison (*orange*).

**Altering the number of firing factors.** We explored the influence of firing factor concentration variations on DNA replication dynamics by manipulating the copy number of firing factors introduced into the model. To isolate the effects of firing factor concentration changes, we maintained the constant recycling rate of 0.05 and kept the origin firing rates which were derived from fitting the baseline model configuration in which $F = 200$. We quantitatively assessed the impact of altering the firing factor copy numbers by computing the MAE in replication timing across a range of values from $F = 25$ to $F = 800$. This data was subsequently fitted a fifth-degree polynomial (Fig 5B). As expected, increasing the number of firing factors led to earlier replication timing and a higher number of origins firing, whereas decreasing the number of firing factors produced the opposite effect. However, the shape of the replication timing curve remained remarkably robust despite this severe perturbation. This is illustrated for chromosome 2 in Fig 5C.

As shown in Fig 5B, successive increases in firing factor copy numbers causes diminishing increases in error as they become less limiting and therefore their availability has less influence on origin firing dynamics. Conversely, reducing the copy number of firing factors incrementally increases the error. This is because, in our model, firing factors are essential for initiating origin firing; therefore, a minimum number is required to ensure sufficient origins can activate to complete DNA replication within the expected timeframe.

## Discussion

Despite being central to maintaining genetic integrity, the mechanisms underlying the temporal regulation of DNA replication remain poorly understood. Since DNA replication is a stochastic process and therefore differs between cells, valuable information is lost in bulk experiments which use population averages [18,19]. Whilst the development of high-throughput, single-molecule experimental techniques have advanced our ability to investigate DNA replication, the complex cellular environment means it is still difficult to identify the features most important for determining its dynamics [21,22,25,46–48].

Here, we created a stochastic model for *S. cerevisiae* whole-genome replication from first principles. By making the model as simple as possible whilst still able to reproduce experimentally determined measures of DNA replication dynamics, we highlight the features which are most important for determining these dynamics. The ability to rapidly conduct thousands of simulations of our model offers a scale of repetition far surpassing what experimental methods can feasibly achieve. Moreover, conventional experimental techniques typically focus on only a few aspects of DNA replication dynamics. In contrast, our model allows numerous features to be calculated from a single output, providing a more comprehensive insight. Although our model depends on replication timing data from [38] to fit origin firing rates, its ability to closely reproduce independent, experimentally determined replication dynamics suggests that it effectively captures broader aspects of replication beyond the dataset used for fitting.

In simplifying the model, we used firing factors, recycling rates, and origin firing rates to combine many complex features affecting DNA replication into a few basic concepts. In our model, firing factors represent the completed pre-IC. The number of available firing factors represents any protein or protein complex essential for forming the pre-IC that may be in limiting abundance at any given time during S phase. For instance, this may represent the availability of the complex formed during pre-IC assembly between Sld2, Sld3, and Dpb11 which are all present in limited abundance [15]. This complex is released before origin firing and therefore does not move with the replication forks [11,12]. Based on this, in our model, firing factor recycling is independent of replication fork termination and, instead, recycling

begins at a set rate immediately following firing. Of the essential firing factors thought to be in limited abundance, only Cdc45 is known to travel with replication forks. Over-expression of Cdc45 alone is insufficient to cause premature firing of late origins, suggesting it is not solely responsible for limiting origin firing [14]. Despite this, the DNA replication model developed by [32] employed a fork-dependent mechanism for recycling firing factors. This assumes that the firing factors travel with the replication forks, only becoming available again once the forks terminate. However, the ability of our model to be fitted and validated without fork-dependent recycling suggests that it is not strictly necessary to model DNA replication dynamics. In our model, recycling rates represent a generic time-lag which encapsulates any processes which may delay the ability of firing factors to activate new origins following their release from a fired origin. This could include the time required for diffusion and the assembly of protein complexes required to construct the replisome. However, by not accounting for differences in the time taken for firing factors to diffuse to different origins, our model does not incorporate the effects of spatial proximity, where origins located near recently fired origins may have increased firing probabilities due to the release of firing factors.

Based on the assumption of a rapid transition from G1 to S phase, in our model, all firing factors are available from the beginning of the simulation. However, although the transition from G1 to S-phase is accompanied by a switch-like increase in CDK and DDK activity, it is possible that the subsequent activation of firing factors required for initiating DNA replication is more gradual [9]. This would be expected to increase the competition for firing factors at the beginning of S phase, thereby decreasing the initial rate of origin firing.

Whilst the number of firing factors and their recycling rates affect the firing probability of all origins uniformly, individual differences in firing probability arise from each origin's specific assigned firing rate. An origin's firing rate encompasses anything which may impact its ability to associate with firing factors, including sequence composition and local chromatin structure. There is evidence suggesting that the probabilities of origins firing are dynamically regulated throughout S phase by mechanisms such as cooperative firing, the formation and dissolution of replication factories, and changes in chromatin environments [16,17,49–52]. However, our model simplifies these dynamics by assuming constant origin firing rates for the entirety of S phase. Furthermore, origin firing rates, recycling rates, and the maximum number of firing factors remain constant between simulations, implying no variability in these parameters between cells or across cell cycles. The ability of our model to reproduce observed DNA replication dynamics without accounting for these variables suggests that such changes are less central to determining the overall dynamics and may instead fine-tune DNA replication. This fine-tuning could enhance replication robustness by allowing origin firing rates to respond to system and environmental changes [53].

Our model also assumes a constant rate of fork movement in order to be consistent with findings from [54] and [25] which both demonstrated a remarkable uniformity in replication fork speeds across the *S. cerevisiae* genome. [25] also observed a slight acceleration of fork speeds during the S phase and identified specific genomic regions, such as centromeres, telomeres, and tRNA genes, where replication forks progress more slowly, likely due to increased pausing and stalling at replication barriers [55,56]. Neither increasing fork speeds nor fork stalling are included in our model. Despite these simplifications, our model's ability to successfully reproduce experimentally determined measures of *S. cerevisiae* DNA replication suggests that such features may not be central to determining the overall dynamics of DNA replication in *S. cerevisiae*. Whilst changing fork speeds and fork stalling could easily be incorporated into our model, the constant rate of fork movement also limits the maximum gradient of the DNA replication timing profile which helps to avoid over fitting. This

constraint may partly explain the poorer fit of the simulated replication timing to experimental data for chromosome 1, which exhibits steep gradients in replication timing (Fig 2A). Although slower fork speeds could potentially improve the fit, such speeds are not supported by experimental data.

Additionally, our model restricts the initiation of DNA replication to specific sites on the genome, sourced from OriDB [6], and does not capture all potential origins, such as every origin in the rDNA array on chromosome 12. Notably, our model does not include every potential origin in the rDNA array on chromosome 12, which consists of approximately 150 tandem repeats, each harbouring a potential origin [57]. This omission may partly account for the slightly higher error observed in our origin efficiency estimates for chromosome 12 relative to experimental data (Fig 3D). Whilst the origins we included are sufficient to reproduce observed DNA replication dynamics, evidence suggests that DNA replication initiation in *S. cerevisiae* is more flexible [24,58], and it is possible that, similar to other eukaryotes, any site could act as an origin. However, some may have much lower probabilities of firing, thereby serving as dormant origins which provide a backup mechanism in case of perturbations [59].

Our model's capacity to predict aspects of DNA replication dynamics which have not yet been determined experimentally offers novel insights and directs future research. For instance, by calculating the number of active replication forks throughout S phase, our model could highlight periods when cells may be particularly susceptible to replication stress or damage. Our model also predicts a non-trivial relationship between the average replication time and the standard deviation of replication time. The inability of our model to infer firing time distributions directly from efficiencies underscores the complexity of origin firing dynamics and suggests that multiple interrelated factors influence these distributions such as the proximity of origins to other origins and their respective efficiencies. Our model also predicted that replication timing curves do not particularly change shape under overexpression and underexpression of firing factors. Although, notably, our model does not consider dNTP availability, which is likely to become limiting following increased firing due to increased firing factor availability. In reality, this would trigger the S phase checkpoint to stabilise replication forks and prevent further origin firing until dNTP levels are restored [60]. Therefore, in this case, increasing the number of firing factors can be interpreted as increasing the overall replication capacity of the cell more generally.

Although our model can be compared to several published approaches, direct one-to-one performance comparisons are challenging due to differences in metrics and assumptions such as whether and how firing rates depend on time or space. Some studies such as [31] focus on the median firing time, rather than replication time, of origins, making side-by-side comparisons with our results less straightforward. Others employ time-dependent origin firing rates [32,61,62], which differ from our model's approach of combining time-independent firing rates with dynamic firing-factor availability. Meanwhile, certain kinetic or neural-network-based models explicitly incorporate a time-dependent probability of origin firing that matches well with our notion that the "effective" initiation rate varies as forks consume a limiting pool of factors, even though our baseline firing rates are time-independent [63]. Furthermore, some authors refine their fits by tracking the width of firing-time distributions or by calculating Pearson correlations of mean replication timing profiles. Although we have not replicated these specific metrics, our model aligns well with experimental data under simpler assumptions, such as fork-independent recycling of firing factors and unrestricted origin firing time distributions. Together, these points clarify that our overall framework is broadly in line with previous work while differing primarily in how we handle the time dependence of initiation and the precise fitting metrics used.

## Conclusion

The simplicity of our model not only identifies the minimal components necessary to reconstruct replication timing profiles, but also shows how the shape of the replication timing curve remains remarkably robust to perturbations such as changes in the number of firing factors, which represent alterations in the global replication capacity of the cell. Our simulation-based model offers substantial advantages in terms of intuitiveness and design flexibility, positioning it as a valuable tool for exploring how various disruptions might influence the dynamics of DNA replication. Given the high conservation of DNA replication mechanisms across species [64], the insights gained from this model are likely generalisable to other eukaryotes.

## Supporting information

**S1 Fig. The Beacon calculus model.** Beacon Calculus code used for the model. Processes and comments are highlighted in *pink* and *green* respectively. The code includes process definitions for firing factors, `FF`, origins, `ORI`, and replication forks, `FR` and `FL`. Comments within the code are indicated by "`\\`". Actions are enclosed within "{}" and are defined as ordered pairs, specifying the action followed by the rate at which it occurs. Handshake communications are denoted by `@factor!` for sending and `@factor?` for receiving on the `factor` channel. Beacon actions are represented by `ch!` for sending, `~ch?` for checking, and `ch?` for receiving, all on the `ch` channel. The values within "[]" following handshake or beacon are transmitted. The code syntax includes "`.`" for sequential statements, "`|`" for parallel statements, and "`+`" for making exclusive choices. Condition gates are represented by "`->`". All origin and firing factor processes are initiated from the beginning of the simulation. However, this has been omitted from this representation for conciseness.
(PDF)

**S2 Fig. Chromosome 2 origin firing time distributions.** Kernel density plot showing the distributions of origin firing times for all origins on chromosome 2. Each line represents a separate origin, with the colour gradient from *blue* to *red* reflecting its efficiency.
(PDF)

## Acknowledgments

This work was performed using resources provided by the Cambridge Service for Data Driven Discovery (CSD3) operated by the University of Cambridge Research Computing Service (www.csd3.cam.ac.uk), provided by Dell EMC and Intel using Tier-2 funding from the Engineering and Physical Sciences Research Council (capital grant EP/T022159/1), and DiRAC funding from the Science and Technology Facilities Council (www.dirac.ac.uk). We thank all members of the Boemo Lab (University of Cambridge Departments of Pathology and Genetics) and Gideon Coster (Institute of Cancer Research) for helpful discussions and comments that greatly improved the manuscript. We also thank the University of St Andrews MBiochem program for their support and guidance.

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
