## [Decision Letter · Decision Letter 0]

18 April 2025

Dear Dr. Boemo,

We are pleased to inform you that your manuscript 'Regulation of replication timing in Saccharomyces cerevisiae' has been provisionally accepted for publication in PLOS Computational Biology.

Best regards,

Vladimir B Teif, Ph.D.

Academic Editor

PLOS Computational Biology

Mark Alber

Section Editor

PLOS Computational Biology

The original reviewers from Review Commons were not available for this round of review, but both new reviewers from PLoS Comp Biol agree that the authors have adequately addressed the comments from the previous reviews from Review Commons and that the manuscript is publishable with minor text adjustments. The authors are encouraged to address the minor suggestions in the most recent reviews when submitting the final version of the manuscript. Given that these will be addressed, the manuscript can be provisionally accepted.

Reviewer's Responses to Questions

**Comments to the Authors:**

Reviewer #1: This manuscript describes the development of an algorithm to model replication timing data in budding yeast. Understating the basics mechanics of how DNA replication proceeds across genomes is complicated by the high degree of variation in replication programs across cells. In this report the Boemo lab has developed a stochastic model which in which origins compete for limiting initiation factors. Their results show that the model works very well and fits experimental data. The model allows the authors to predict the number of replication forks active at a given time in S phase and other important aspects of DNA replication.

My only criticism of the paper is that it is unclear whether the models of limiting or increasing initiation factors are correct e.g. fig5. The limited experimental data in these conditions (e.g McGuffey Mol Cell 2013) indicates that over expression of limiting factors significantly increases origin efficiency and may flatten the profile more than what is observed in the model. It would be useful if the authors could discuss this apparent discrepancy in the text.

Reviewer #2: In this study, Berners-Lee et al present an elegant and simple stochastic model where the authors distill down elements involved in origin firing to a general ‘firing factor’ and use this as a major parameter for modeling replication timing. The model appears to fit the data very well using only these factors. This model will be of interest to both computational and experimental biologists.

Major comments:

The authors adequately addressed the previous reviewers in their revision plan and updated manuscript.

Minor comments:

While the Github page is relatively easy to follow, it would be good to polish the Github page and documentation a bit for a wider user base (e.g. there is a small typo in the readme, ‘limtted firing’ should presumably be ‘limited firing’).

The final sentence is a bit of an overreach. While there are many conserved mechanistic features across eukaryotes, the divergence is likely to be too great for the current model to be used (e.g. mammalian origins don’t have defined sequences). I would emphasize the usefulness of the model for answering fundamental questions in the most distilled system possible, which is still beneficial for scientific progress.

[only a suggestion]: To further boost the reach of this model to experimentalists, the authors might consider constructing a simple webapp (e.g. in Shiny for python or Dash) where users can implement changes of interest and see how results might compare to their datasets.

**Have the authors made all data and (if applicable) computational code underlying the findings in their manuscript fully available?**

Reviewer #1: None

Reviewer #2: Yes

PLOS authors have the option to publish the peer review history of their article (what does this mean?). If published, this will include your full peer review and any attached files.

Reviewer #1: No

Reviewer #2: No

---

## [Editor Report · Acceptance letter]

PCOMPBIOL-D-25-00489

Regulation of replication timing in Saccharomyces cerevisiae

Dear Dr Boemo,

I am pleased to inform you that your manuscript has been formally accepted for publication in PLOS Computational Biology. Your manuscript is now with our production department and you will be notified of the publication date in due course.

With kind regards,

Zsofia Freund
